# Stochastic generation of spatially coherent river discharge peaks for continental event-based flood risk assessment.

Dirk Diederen[1], Ye Liu[1], Ben Gouldby[1], Ferdinand Diermanse[2], and Sergiy Vorogushyn[3]

[1]HR Wallingford, Crowmarsh Gifford, United Kingdom
[2]Deltares, Delft, Netherlands
[3]GFZ German Research Centre for Geosciences, Potsdam, Germany

*Correspondence to:* Dirk Diederen (dirkdiederen@hotmail.com)

**Abstract.** We present a new method to generate spatially coherent river discharge peaks over multiple river basins, which can be used for continental event-based probabilistic flood risk assessment. We first extract extreme events from river discharge time series data over a large set of locations by applying new peak-identification and peak-matching methods. Then we describe these events using the discharge peak at each location, whilst accounting for the fact that the events do not affect all locations. Lastly we fit the state-of-the-art multivariate extreme value distribution to the discharge peaks, and generate from the fitted model a large catalogue of spatially coherent synthetic event descriptors. We demonstrate the capability of this approach in capturing the statistical dependence over all considered locations. We also discuss the limitations of this approach and investigate the sensitivity of the outcome to various model parameters.

## 1 Introduction

Flood events cause large damages worldwide (Desai et al., 2015). Flood risk assessments (FRAs) are required for long-term planning, e.g. for investments in infrastructure and other urban capital. Following the definition of risk (Field, 2012), simply put as probability of damage, FRA requires an approximation of the risk curve under stationary climate conditions and a current distribution of asset values. Typically, for FRAs a chain of models is applied, covering the entire risk cascade from hazardous extreme events down to flood damages or casualties resulting from inundation (e.g. expected annual damage, loss of life). The risk curve represents the probability of damages and is approximated by the evaluation of a comprehensive catalogue of hazard scenarios. The chain can be run in continuous mode (Cameron et al., 1999; Boughton and Droop, 2003; Borgomeo et al., 2015; Falter et al., 2015), or with separate events (Vorogushyn et al., 2010; Gouldby et al., 2017). To drive the chain of models, boundary forcing is required. This typically comprises a large catalogue of synthetic forcing data, with models conditioned on observations.

Widespread flooding can potentially cause large damage in a short time window. Continental events and, for instance, maximum probable damages are of interest. In particular, the (re)insurance industry wants to know the chance of a widespread

portfolio of assets getting affected in a short time window (Jongman et al., 2014). With the increase in computational power, continental FRAs have recently become feasible (Ward et al., 2013; Alfieri et al., 2014; Dottori et al., 2016; Vousdoukas, 2016; Winsemius et al., 2016; Paprotny et al., 2017; Serinaldi and Kilsby, 2017). Vorogushyn et al. (2018) call for new methods for large FRA to enable the capturing of system interactions and feedbacks. The focus in this study is on methodology required
for the generation of a large catalogue of synthetic continental discharge event descriptors for fluvial FRA.

     River discharge waves may cause the exceedance of bankful conditions or may cause dikes to fail. They are dynamic, i.e. show a wave-like behaviour. Travel times of discharge waves in large river basins can be long, i.e. time lags between discharge peaks at different locations can be large. With large travel times, a new discharge wave may be generated upstream, while the previous discharge wave has not yet reached the river mouth. Furthermore, discharge waves in river basins are triggered by
atmospheric events that may span across multiple river basins. Finally, discharge waves in different river basins may be related to a single atmospheric event, but do not occur at the same time, since catchments have different response times. With an increasing spatial domain, dynamic events start overlapping in time and merge into a space-time continuum. For a continental FRA, the challenge arises how to define observed continental river discharge events and how to simulate synthetic continental river discharge events while retaining the observed statistical properties in space (spatial dependence/coherence).

We distinguish between two groups of event identification methods: methods based on time blocks and methods based on dynamic events. Using blocks, events are defined within fixed time windows and described by their statistical properties, e.g. annual maximum discharge. The main advantage of the blocks method is its simplicity, allowing statistical properties to be rapidly captured. Dynamic events are defined as events with spatially varying time windows, which are based on the discharge values. As described above, for large spatial domains small dynamic events at different locations may overlap in time and form
one single long-lasting spatio-temporal event. Hence, a practical definition of dynamic space-time windows is required.

     We analysed pan-European discharge waves in the space-time continuum, which are characterised by significant time lags between peaks at distant locations. We applied a new method of dynamic event identification where we aimed to capture discharge events in each major European river basin, after which we used a block-based time window method to merge them to spatially-coherent, pan-European events. We described the pan-European discharge events by their peaks with which we
parameterised a stochastic event-based generator of event descriptors. Using the generator, we simulated synthetic descriptor sets, after which we compared the statistical properties of the synthetic sets to those of the observed. Finally, we discussed the main limitations of the methodology and the choice of parameter settings.

## 2   Methodology

### 2.1   Observed data

We used a gridded discharge reanalysis data set covering major river networks in Europe, which was obtained with the well-established LISFLOOD model (Van Der Knijff et al., 2010). This data set resulted from a hydrological model driven by a climate reanalysis data set for the period 1990 to 2015. It has a spatial resolution of 5x5 km and a daily temporal resolution. A high temporal resolution is critical for river discharge waves to be tracked in the extended river network.

In order to keep the computational costs reasonable, the network was reduced to the major streams and tributaries. This means that, although the input data was two-dimensional in space $(x,y)$, we considered the network of 1-dimensional rivers $(s)$. For high-order small streams to be included, a higher spatial and temporal resolution would be required for wave tracking. Although the data was derived from a modelled reanalysis data set, we refer to the used subset of data as 'observed data' as

it comprises observations of reality, contrasting with 'synthetic data' which comprises data values of what may hypothetically occur.

## 2.2    Objectives, framework and quality check

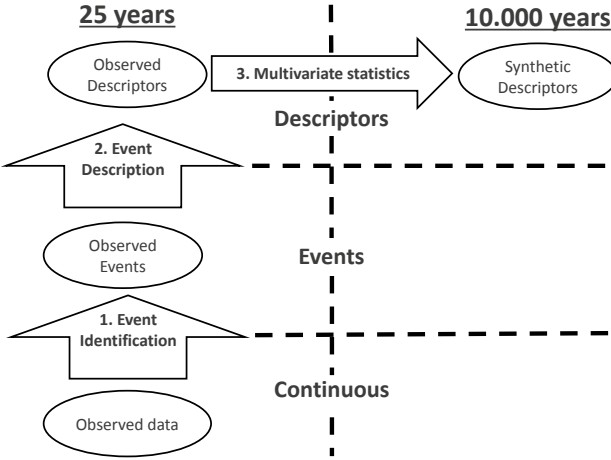

**Figure 1.** The applied framework, which comprises three steps. First, events are identified in the observed data. Second, the observed events are described, providing a matrix of observed descriptors. Third, multivariate statistics is applied to generate a large matrix of coherent synthetic descriptors.

In this study there were two objectives. First, to capture the spatial dependence structure between peaks of discharge events at different locations spread out through Europe (OBJ1). Second, to generate a large catalogue containing synthetic discharge

peaks, filling up the observed distributions while retaining the observed dependence structure (OBJ2).

The framework for the generation of synthetic peak sets consisted of three consecutive steps, see Fig.(1). First, the identification of continental events in the continuous data on the entire pan-European river network (OBJ1). To achieve this, we started by identifying local events (single location), for which we applied a new method of time series analysis, 'Noise Removal' (NR), at every location (grid cell) in the river network. These local events were connected to neighbouring locations

to obtain river basin events, to be subsequently merged to pan-European events, which span across multiple river basins. Second, the description of the pan-European events (OBJ1). To reduce the dimension (number of locations) for statistical analysis while trying to maintain an acceptable spatial coverage, we selected 298 representative locations within the network of major European rivers, see Fig.(2). At these representative locations, we described the continental discharge events by their local

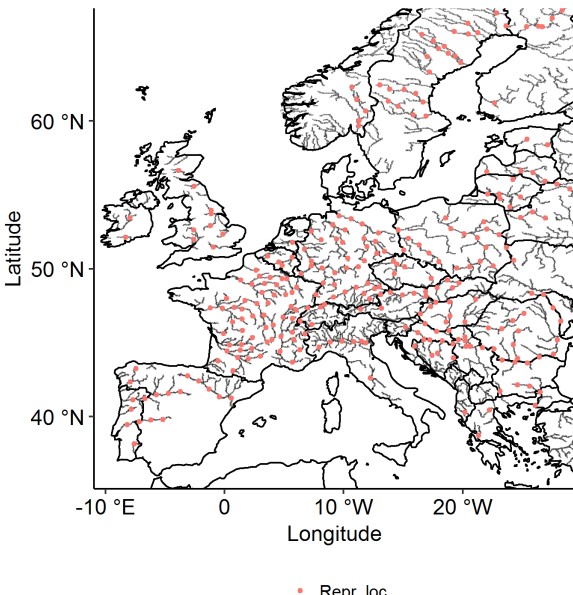

**Figure 2.** The network of major European rivers and a subset of 298 representative locations.

peaks. Third, the generation of a large catalogue of synthetic descriptor using multivariate statistical analysis (OBJ2). We fitted a multivariate dependence model to the catalogue of observed descriptors covering 25 years, retaining the observed spatial correlation structure. Finally ,the fitted statistical model was used to simulate a large catalogue of synthetic event descriptors, characterised by spatial coherence and comprising a synthetic period of 10,000 years.

5    We considered the following as the key features for the quality of the generated catalogue of synthetic event descriptors. First, it should contain descriptions of a much larger variety of hypothetical (synthetic) events than the events identified in the observed data (KF1). Second, the dependence structure of the synthetic catalogue needs to agree with that of the observed, since the catalogue of observed event descriptors should be a likely subset of the synthetic catalogue (KF2).

## 3   Events

### 3.1   Single-location events

When using the popular 'Peaks-Over-Threshold' method (POT) per location, all events below a particular threshold are dropped. This is appropriate for event identification only when events show a homogeneous 'extremeness-per-location'. However, when studying discharge waves moving through the river network by 'extremeness-per-location', a heterogeneous behaviour can be expected. Relatively extreme events upstream may become less extreme moving downstream when the lower 15   part of the river basin is not activated. Or, in contrast, relatively non-extreme events at different upstream branches can generate

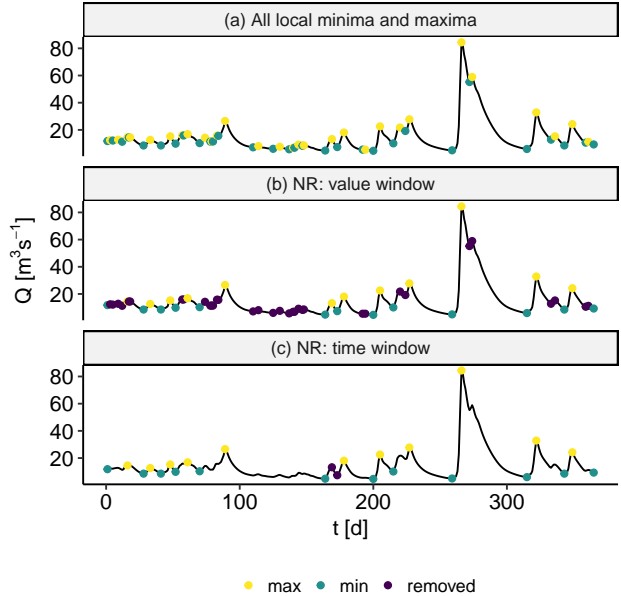

**Figure 3.** (a) All local minima and maxima. (b) Removal of noise using the value window. (c) Removal of noise using the time window.

a relatively extreme event at confluences downstream due to wave superposition. To address the heterogeneity, we developed a new noise removal algorithm to capture local events which manages to eliminate small local peaks that are part of a bigger event (noise), while retaining small events that may be spatially connected to larger events upstream or downstream. This is a key feature to the wave tracking, which will be introduced in Sect.[3.2].

The procedure of NR is as follows. First, all local minima $\mu$ and maxima $M$ are identified, defined as the points where the sign of the increment changes from positive to negative and vice versa, see Fig.(3a). Second, small perturbations are identified as noise and are removed, where the following algorithm is applied:

1. Define a series $Y = (\mu_1, M_1, \mu_2, .., \mu_{n-1}, M_{n-1}, \mu_n)$ and calculate $dY = |\mu_1 - M_1, M_1 - \mu_2, .., M_{n-1} - \mu_n|$.

2. Either calculate the 'NR value window' $\delta_y = f_y \times max(dY)$, where $f_y$ is a fraction to set, or set $\delta_y$ directly.

3. Find $i$ by selecting the smallest difference in value $dY_i = min(dY)$. If $dY_i < \delta_y$, remove $Y_i$ and $Y_{i+1}$ from $Y$, then recalculate $dY$. This step is repeated until there is nothing left to remove.

An example of the NR value window filtering is displayed in Fig.(3b). Third, to make sure that two local minima are not too close in time, the following algorithm is applied:

1. Define a series $T = (t_{\mu_1}, t_{\mu_2}, .., t_{\mu_n})$ and calculate $dT = \left(t_{\mu_2} - t_{\mu_1}, t_{\mu_3} - t_{\mu_2}, .., t_{\mu_n} - t_{\mu_{n-1}}\right)$.

2. Either calculate the 'NR time window' $\delta_t = f_t \times max(dT)$, where $f_t$ is a fraction to set, or set $\delta_t$ directly.

3. Find $i$ by selecting the smallest difference in time $dT_i = min(dT)$. If $\mu_i < \mu_{i+1}$, $j = i + 1$, else $j = i$. If $M_{j-1} < M_j$, $k = j - 1$, else $k = j$. Remove $Y_{2j-1}$, $Y_{2k}$ and $T_j$, then recalculate $dT$. Repeat this step until there is nothing left to remove.

An example of the NR time window filtering is displayed in Fig.(3c). Fourth, a local event can be chosen to last from minimum to minimum, or to be only the time step in which the peak occurs, or something in between.

We set the NR value window fraction relatively low $f_y = 0.01[-]$, such that many small local events were retained. However, by setting the fraction low, small perturbations (noise) made it difficult to spatially separate events. This was ameliorated by using the NR time window $\delta_t = 10d$, ensuring a minimal amount of time between local minima. The choice of NR parameters will be elaborated in Sect.[5.2.1].

## 3.2 River basin events

River discharge waves propagate through the network in downstream direction, introducing time lags between the moments the waves pass at different locations. Time lags are difficult to estimate, because the celerity of river discharge waves can be highly nonlinear. The wave celerity is a function of the hydraulic depth and changes in a nonlinear way when overbank flow occurs and floodplains become inundated. When using gauge data (point-observations), combining local events to events that span multiple locations, time lags are typically addressed using time windows. The gridded data set used in this study allowed us to try a new method to combine local events to river basin events, which we refer to as 'wave tracking'. Each location in the river network is physically connected to its neighbouring locations, which allows waves to be tracked throughout the entire river network. Wave tracking is robust to non-linearities in the wave celerity, and therefore it allows to better address time lags, so that, when we compare peaks at different locations in Sect.[4], we make sure they are of the same discharge wave.

To track river discharge waves, we applied the following procedure. First, we separated local events by applying NR to time series at every location in the river network, where of each local event we retained the day of the peaks $\pm 1$ day. Second, we identified separate events per river branch by capturing the polygons in the branch's space-time image, see Fig.(4). The settings of the NR were adjusted by trial and error to try to obtain consistent polygons in space (low noise removal), but separated in time (high noise removal). Third, we merged the events of different river branches based on overlap of event time coordinates at the confluences. This procedure resulted in a variable number of tracked discharge waves per river basin.

## 3.3 Pan-European events

Precipitation events, which are the main driving source of river discharge events, span across different river basins. Therefore, large discharge events in adjacent river basins are likely to be correlated. To account for this correlation, we had to define events that included discharge waves across different river basins (in this study pan-European events). Since discharge waves do not span across different river basins (by definition), such events should be connected to each other in a different way. Discharge waves in different basins are not synchronised, which adds additional complexity. In order to obtain a method to construct

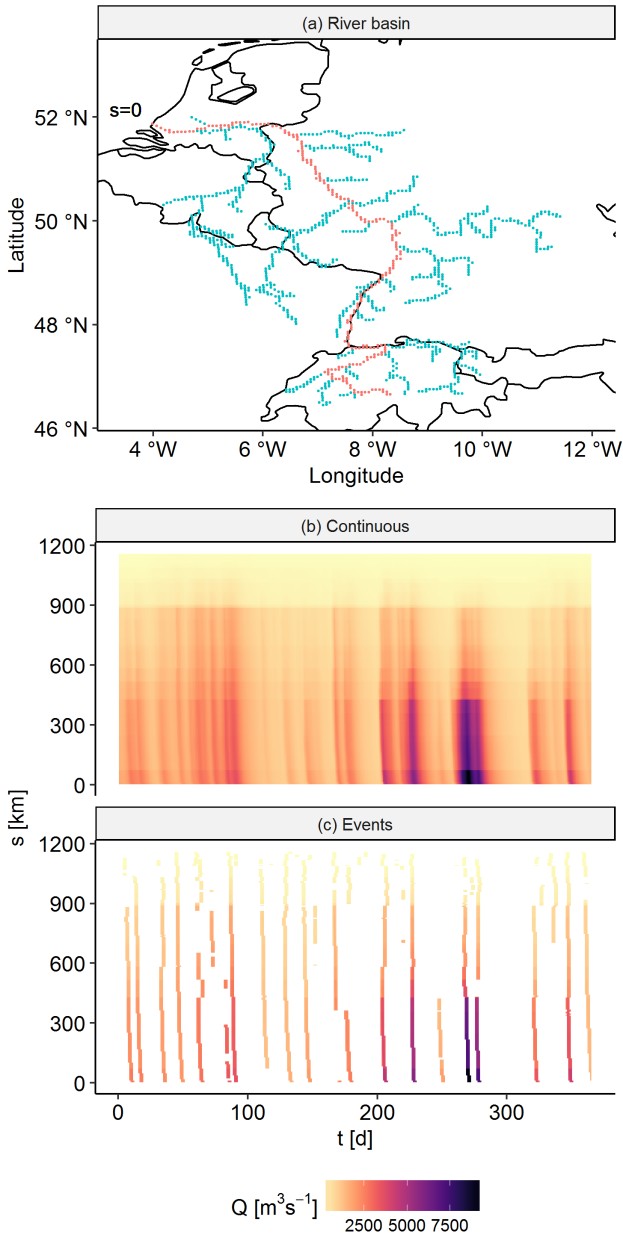

**Figure 4.** a) A particular branch of the river Rhine. b) The continuous discharge data on the river branch, where the river mouth is located at $s = 0km$ and the head water is located around $s = 1100km$. c) Events on the river branch. The polygons (i.e. separated islands of data) are discharge waves moving through the river branch.

pan-European events, which on the one hand considers discharge waves in river basins and on the other hand accounts for trans-basin dependence, we propose a combined approach of wave tracking and 'global time windows'.

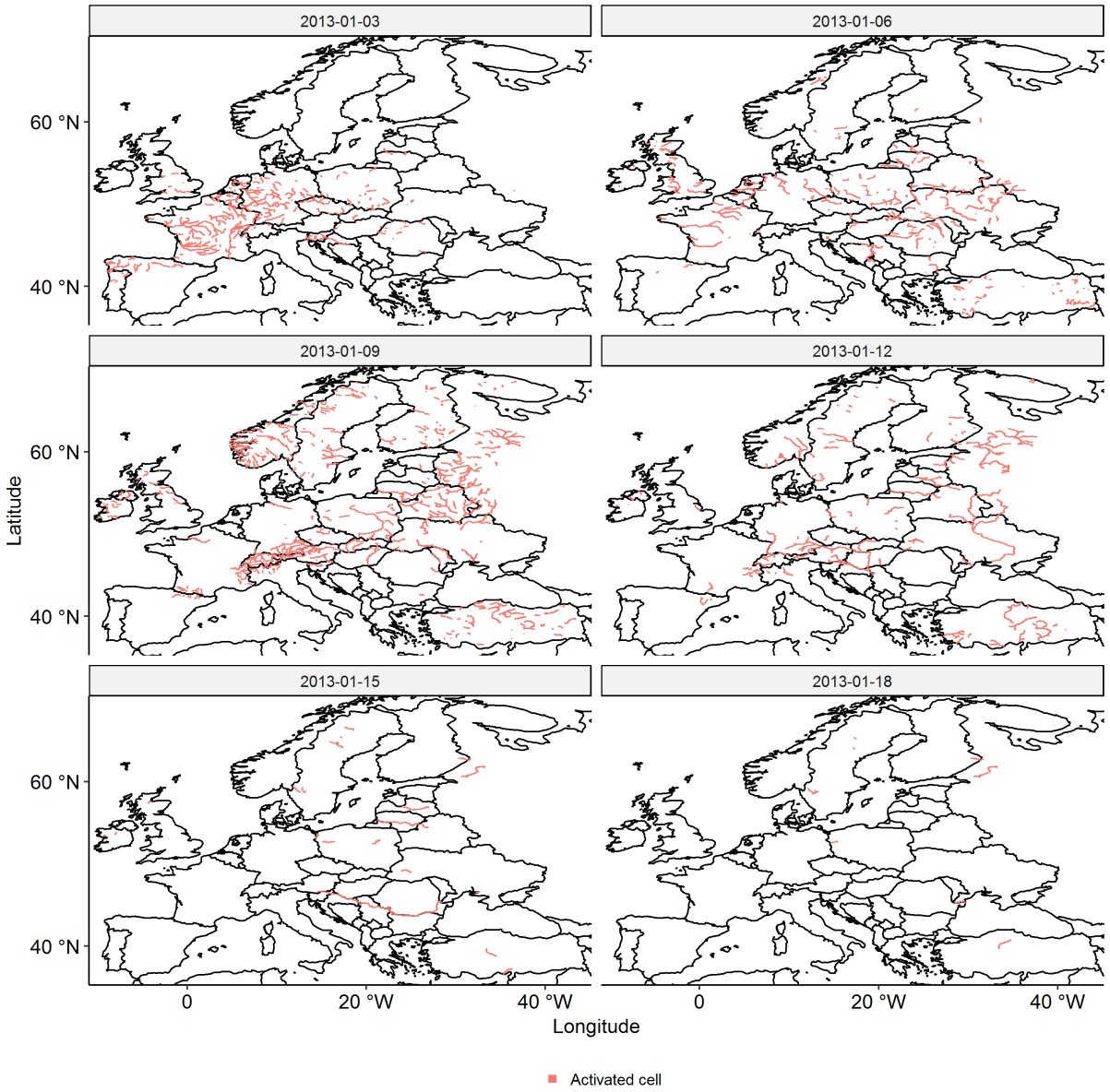

**Figure 5.** Daily snapshots of a Pan-European event with a large spatial extent.

The following procedure was adopted. First, we set up subsequent global time windows with a length of $\delta_l = 21d$, which resulted in 428 global time windows in the period 1990-2015 (i.e. 428 pan-European events). We will discuss the length of the global time windows in Sect.[5.2.1]. Second, to each global time window we assigned complete, tracked discharge waves. To do this, we let each discharge wave be represented by its first time coordinate, i.e. the day when the discharge wave started somewhere (upstream) in the river basin. The discharge wave was then assigned to the global time window in which this day

occurred. If, per river basin, multiple discharge waves were assigned to a particular global time window, we only retained the discharge wave with the largest discharge value. This procedure resulted in 428 pan-European events. An example of a pan-European event is displayed in Fig.(5).

## 3.4  Event description

We aimed to describe the pan-European events by their peak discharge, at 298 representative locations on the river network. However, the pan-European events did not yield discharge peaks at all representative locations for each event, i.e the observed descriptor matrix had gaps. To be able to capture the spatial dependence structure in Sect.[4], we had to fill the gaps by assigning 'auxiliary values'. This will be further discussed in Sect.[5.2.1].

We applied the following procedure. At locations where an event occurred, we extracted the discharge peak. Where no event

occurred (36% of the entries in the observed descriptor matrix), we filled the gaps using auxiliary values. Per representative location (i.e. column-wise) we set up a number of local time windows in between the peaks of identified events, corresponding to the number of gaps between those respective peaks. Within each of these local time windows, we selected the maximum value as auxiliary value. This procedure resulted in a (complete) observed descriptor matrix.

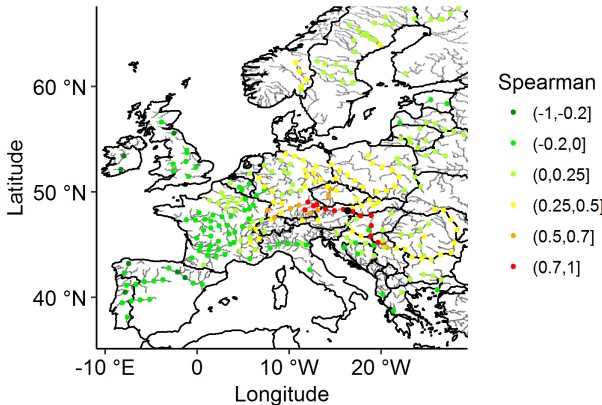

**Figure 6.** Correlation of descriptors at all representative locations versus descriptors at Vienna (black dot).

Fig.(6) shows the correlation of the descriptors with those at Vienna. Good agreement was found with the work of Jongman

et al. (2014). It can be observed that the highest correlation was found at nearby locations within the same river basin. However, significant correlation was found in nearby locations that were not in the same river basin, which confirmed the importance of identifying events spanning multiple river basins.

In order to align with the corresponding literature in statistical models for multivariate extreme values, in the next Sect.[4] columns of the observed descriptor matrix will be referred to as margins and the large values in each column will be referred

to as the upper tails of the marginal distributions.

## 4 Multivariate statistical model

### 4.1 Marginal distributions

We fitted Generalised Pareto Distributions (GPDs) (Coles et al., 2001) to the upper tail of the marginal distributions, i.e. for each column in the observed descriptor matrix. The issue of threshold choice for GPDs is well-discussed in the literature (Northrop et al., 2017). After comparing the model fits, we used the $\zeta_m = 0.94$ empirical quantile as marginal threshold for the GPD at each location. This threshold was found by trial and error, which will be elaborated in Sect.[5.2.2]. We tested the quality of the marginal GPD fits with a standard method, comparing the empirical quantiles and probabilities against the modelled, including checks of the tolerance intervals.

### 4.2 Multivariate dependence model

To be able to capture the dependence between sets of descriptors (i.e. rows in the observed descriptor matrix), we started by transforming the marginals to the uniform space. This transformation is applied in many other analyses, e.g. copulas (Genest and Favre, 2007; Nelsen, 2007). Values below the marginal threshold used to fit the GPDs in Sect.[4.1] were transformed using the empirical marginal distribution and values above the marginal threshold were transformed using the GPDs. We applied a model with two different components to capture the dependence structure, one for the non-extreme part and one for the extreme part.

The dependence structure of the non-extreme part was captured using a non-parametric, multivariate kernel density model with Gaussian kernels. We transformed the (entire) uniform marginals to the normal space, with the mean $\mu = 0$ and the standard deviation $sd = 1$. Bandwidths for the kernels where selected using the method of Silverman (2018).

To capture the dependence of the extreme part we chose the model of Heffernan and Tawn (2004), hereafter referred to as 'HT04'. HT04 is a pair-wise dependence model that can be described as a method of lines, $Y_i = aY_{-i} + Y_{-i}^b Z$. Two HT04 model fits are required for each pair of marginals, with either marginal as the conditioning marginal $Y_i$ and the other as the dependent marginal $Y_{-i}$. Each fit holds two parameters, $a$ and $b$, after which a residual $Z$ is calculated from each observed data point. The data used to fit the model are the pairs where the conditioning marginal $Y_i$ is larger than a fitting threshold $\zeta_f$. With an infinite number of samples drawn from HT04, each model fit would result in as many pair-wise lines as there are data points. However, for simulation a subset of these lines is used, since HT04 should be applied only if the largest marginal in the set is above a particular simulation threshold .

To fit HT04, we transformed the (entire) uniform marginals to the Laplace space (Keef et al., 2013). We obtained HT04 model fits in the Laplace space using maximum likelihood, with each marginal as conditioning variable and all other marginals as dependent variables, resulting in a total of 298*297 model fits, where we chose the fitting threshold $\zeta_f = 0.9$, which was a trade-off between variance and bias. HT04 was recently applied for fluvial flooding (Keef et al., 2009; Lamb et al., 2010; Schneeberger and Steinberger, 2018) and for coastal flooding (Wyncoll and Gouldby, 2015; Gouldby et al., 2017), in which the model fitting procedure is described in more detail.

## 4.3  Simulation

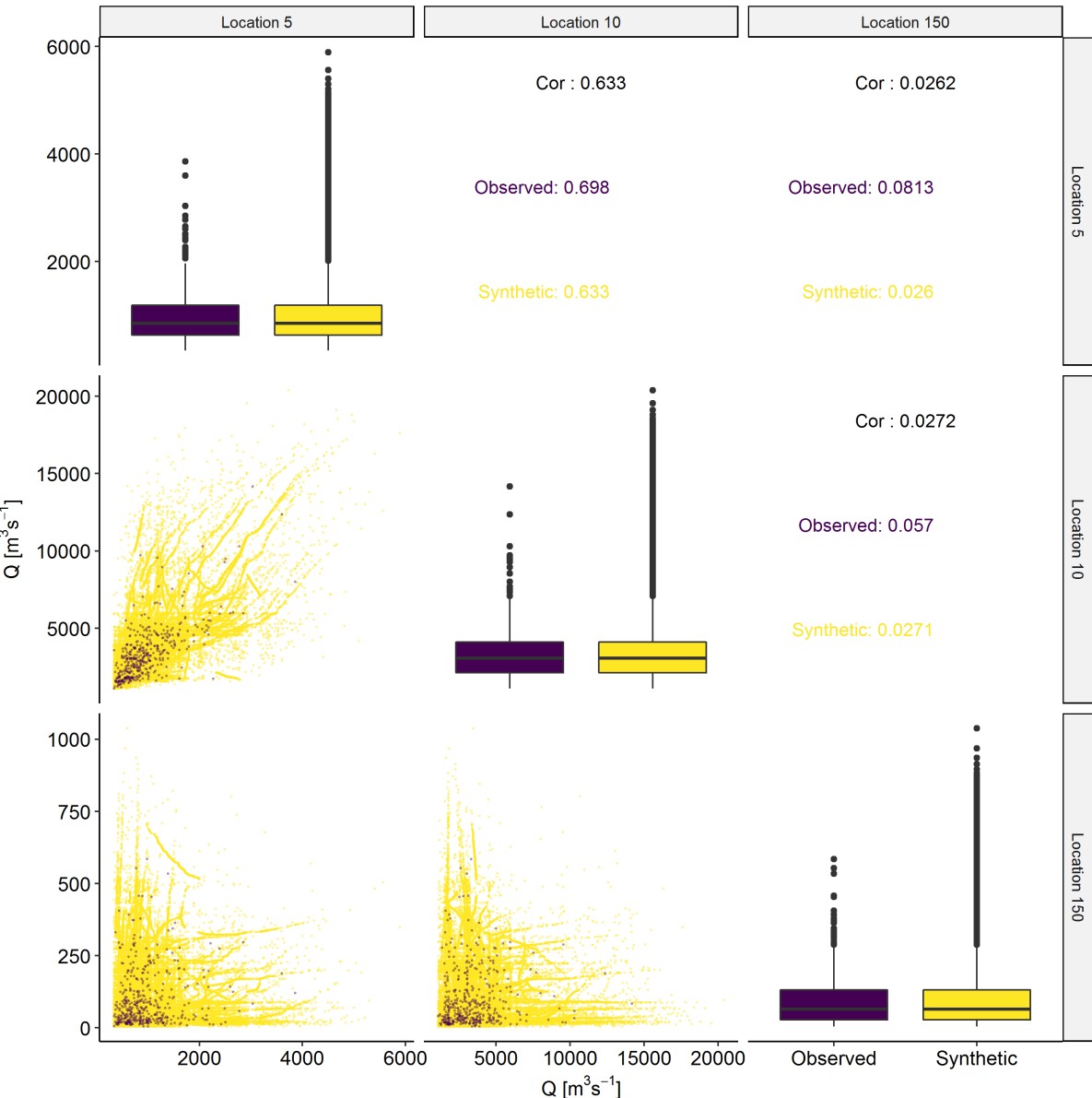

**Figure 7.** Observed (purple) versus synthetic (yellow) descriptors at three locations. In the diagonal, distributions of observed and synthetic descriptors per location are compared using box-plots. Below the diagonal, pair-wise scatter plots are displayed. Above the diagonal, pairwise correlations are displayed.

We split the observed uniform descriptor matrix into a 'non-extreme' part, and an 'extreme' part. Each row in which not a single descriptor exceeded an extremal simulation threshold $\zeta_e = 0.98$ was determined to be non-extreme (23%), the rest (77%) was determined to be extreme (somewhere). For the non-extreme sets, we re-sampled from the non-parametric model. For the extreme sets, we re-sampled from HT04, where the model fit was used of the marginal that was the largest by quantile in the set. All sets were re-sampled $N = T_{sim}/T_{obs}$ times, where $T_{obs}$ is the duration of the observed data (25 years) and $T_{sim}$ is the duration of the synthetic data (10.000 years). After the simulation, we transformed the marginals of the synthetic descriptor matrix to respect the fitted GPDs. This implies that we forced the synthetic marginals to have the same distribution as the corresponding observed marginals. However, by forcing this transformation we slightly distorted the dependence structure.

### 4.4 Quality of the synthetic descriptor catalogue

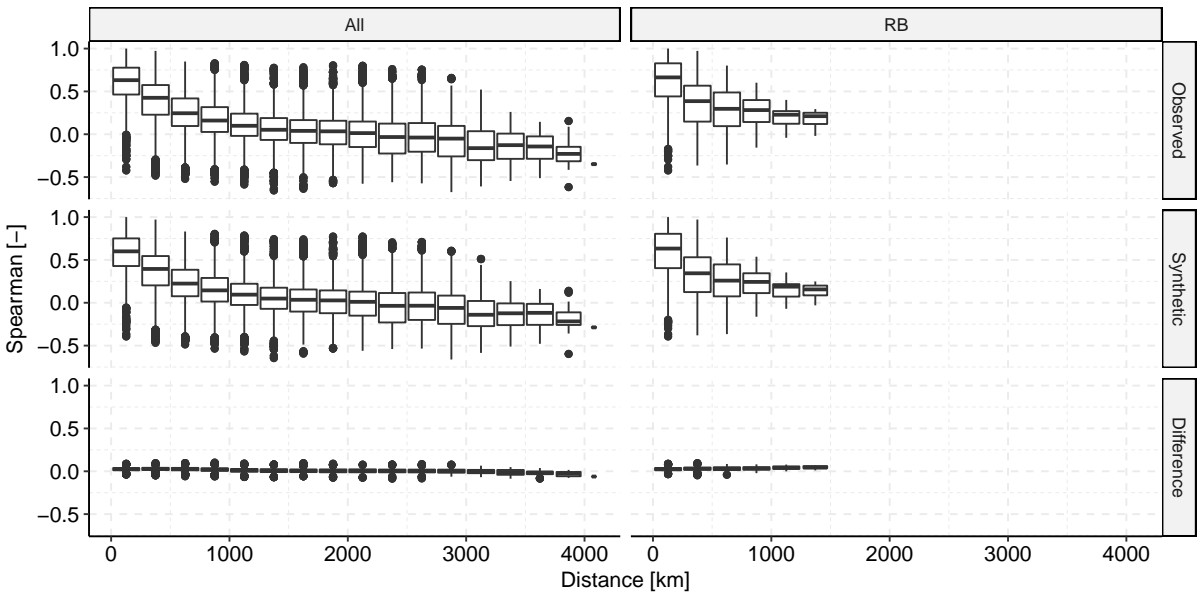

**Figure 8.** Spatial correlation of the observed descriptors versus the synthetic descriptors, summarised by pair-wise Spearman correlation. The upper panel shows the correlation in the observed data, the middle panel shows the correlation in the synthetic data and the lower panel shows the difference between the observed and the synthetic correlation for each pair. The left column shows the correlation between all pairs of locations, right shows only the pairs that are in the same river basin.

Using multivariate extreme value analysis, we extended the observed descriptor matrix with synthetic data, obtaining a (large) synthetic descriptor matrix. The patterns in the larger synthetic descriptor matrix had to match the patterns found in the smaller observed descriptor matrix. We focused on two main patterns; marginal distributions (a column-wise pattern) and dependence structure (a row-wise pattern). To respect the fitted marginal distributions and, simultaneously, retain the dependence structure is challenging. There is no perfect method for these two objectives. We chose to respect the distributions

fitted to the observed marginals, for which we transformed the synthetic marginals to follow the corresponding observed distributions, as described in Sect.[4.3]. Therefore, we had to compare the dependence structure in the synthetic data with the observed.

To further investigate the dependence structure, Fig.(7) shows a sample of the observed descriptors versus the synthetic descriptors. It can be observed that we managed to fill the distributions of the individual descriptors well (KF1) while retaining the observed dependence structure reasonably well (KF2), as the simulated descriptors follow the trends in the observed data.

Fig.(8) shows the pair-wise, spatial correlation structure between descriptors at different locations. Rather than choosing the distance between locations along the river branch, we chose geospatial distance such that we could compare locations not only within river basins, but also across different river basins. The Spearman correlation coefficients of the observed descriptors and the synthetic descriptors agree very well (KF2), which indicates that the general spatial dependence structure is similar in the observed descriptor matrix and in the synthetic descriptor matrix. The difference indicates an overall slightly higher (positive or negative) correlation in the observed descriptor matrix. A shift from positive to negative correlation can be observed around $2000 - 2500 km$, which may be related to large atmospheric patterns.

Following up on the general check of correlation between the entire descriptor sets, we specifically checked if we managed to capture the tail-end correlations. Fig.(9) shows that the general behaviour of co-occurrence of extremes was relatively well captured in the dependence model (KF2). The general pattern in the synthetic descriptors is reasonably similar to the pattern in the observed descriptors. A small positive bias can be observed, which shows that the dependence model slightly underestimated the frequency of joint occurrence of extremes. The zero difference generally falls within the lower quartile. Moreover, the higher the quantile for which we checked the exceedance, the fewer quantile exceedances to count, which lead to a larger spread in the difference between the observed and the synthetic.

## 5  Discussion

### 5.1  Limitations

**Table 1.** $L_1$-$L_6$ are different locations. Sets 1-4 describe a discharge event. Generally, dynamic discharge events do not occur at all locations, such that peaks (P) cannot be identified for all locations. Therefore, auxiliary values (A) have to be used to fill in the gaps.

|       | $L_1$ | $L_2$ | $L_3$ | $L_4$ | $L_5$ | $L_6$ |
|-------|-------|-------|-------|-------|-------|-------|
| Set 1 | P     | P     | P     | P     | A     | A     |
| Set 2 | A     | P     | P     | A     | P     | A     |
| Set 3 | P     | A     | P     | P     | P     | A     |
| Set 4 | P     | P     | A     | P     | P     | P     |

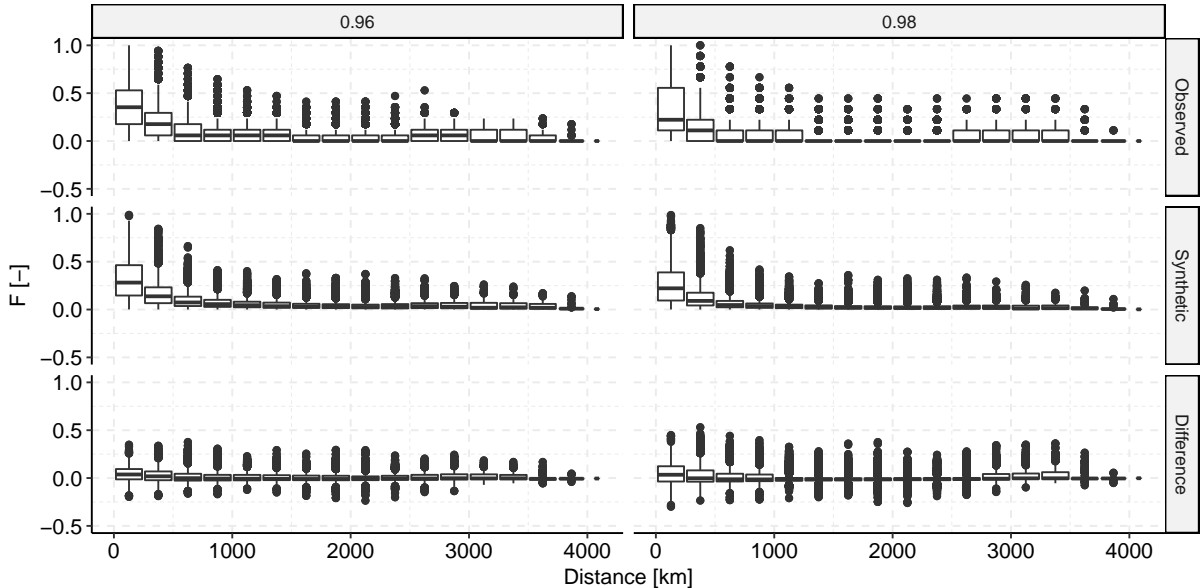

**Figure 9.** Spatial extremal correlation of the observed descriptors versus the synthetic descriptors. For a selection of high quantiles we counted the fraction $F$ of events where extremes at both locations exceeded the respective quantile divided by the total number of quantile exceedances. The upper panel shows the fractions in the observed data, the middle panel shows the fractions in the synthetic data and the lower panel shows the pair-wise difference between the observed and the synthetic fractions.

Historically, observations have been made at specific locations, e.g. discharge gauge stations at certain locations along rivers. Therefore, most event identification methods are designed for local frequency analysis of discharge waves, starting with the identification of 'local events', i.e. events at certain locations, based on temporal dynamics (Tarasova et al., 2018). When addressing spatial dependence using an event-based approach, the difficulty arises that discharge waves will not occur at all

gauged locations within a reasonable time window. The larger the spatial domain in which discharge waves are considered, the more likely it is they are spread out in time. Therefore, an extraction of a dynamic event from a space-time continuum, trying to obtain local peaks for all locations, will lead to a matrix of incomplete peak sets. This is problematic, because current statistical methods for multivariate event generation cannot handle a matrix with missing components (Keef et al., 2009). Therefore, 'auxiliary values', i.e. values that do not represent discharge peaks, are required in order to fill up the gaps (see

Table 1). Different methods exist to assign auxiliary values, for different purposes. Gouldby et al. (2017) analysed different coastal flood variables with an event-based approach, where they adopted concurrent values at all locations where particular thresholds had not been exceeded (i.e. no local event). Keef et al. (2009) relaxed the time constraint, where they considered the values at all locations within a -3 to +3 days time window. Since we were dealing with a large number of locations and with large time lags, neither of these methods were appropriate. Therefore, we found auxiliary values using local time windows,

where these time windows depended on the gaps per location.

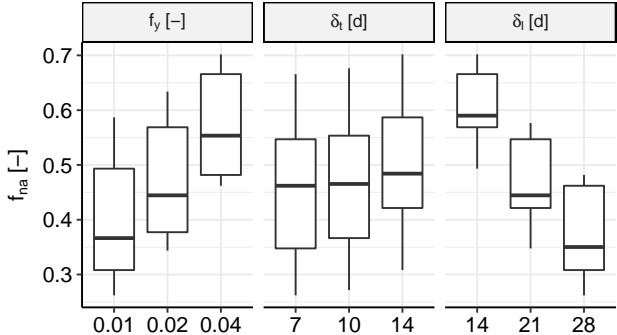

**Figure 10.** Sensitivity of $f_{na}$, which is the fraction of missing values in the observed descriptor matrix, to the three main settings used for the identification of events in Sect.[3]. $f_y$ is the NR value window fraction, $\delta_t$ is the NR time window, $\delta_l$ is the length of the global time windows.

## 5.2 Sensitivity and uncertainty

### 5.2.1 Events

In Sect.[2.2], we stated the objective to capture the spatial dependence structure between peaks of discharge events at different locations spread out through Europe (OBJ1), for which events had to be identified. In the event identification procedure, we
used three main parameters for the identification of pan-European events: the value window fraction $f_y$ and time window $\delta_t$ of the noise removal method (Sect.[3.1]), and the length of the global time windows $\delta_l$ for the pan-European events (Sect.[3.3]). We were dealing with the following trade-off. For each pan-European event, one discharge peak could be assigned to each location. However, depending on the length of the global time window, there may be no river basin event to assign to the global time window, i.e. a missing discharge peak, or multiple river basin events may be assigned from which only one discharge
peak could be retained per global time window per location. Therefore, a relatively large global time window lead to the underestimation of the frequency of discharge waves in river basins, whereas a relatively small global time window lead to a large percentage of missing local events at the representative locations. Since we were dealing with a continental analysis, the fraction of missing values in the observed descriptor matrix $f_{na}$ was relatively large and therefore decisive for our choice of parameter settings.
Fig.(10) shows the sensitivity of $f_{na}$, which is the percentage of missing peaks in the observed descriptor matrix (Sect.[3.4]). When more noise was removed, i.e. larger $f_y$ and $\delta_t$, events had a larger fraction of missing peaks, i.e. larger $f_{na}$. In contrast, a larger $\delta_l$ lead to a smaller fraction of missing peaks $f_{na}$, since the chance was larger for an event to occur at a particular location given more time. When comparing the sensitivity of the three parameters, it can be observed that the outcome is relatively stable with regard to the choice of $f_y$ and $\delta_t$, whereas the percentage of missing peaks $f_na$ could vary quite a lot with
$\delta_l$. Our final choices are $f_y = 0.01[-]$, $\delta_t = 10d$ and $\delta_l = 21d$. A lower $\delta_l$ would have caused too many peaks being missing, which would have lead to unreliable estimation of the dependence model.

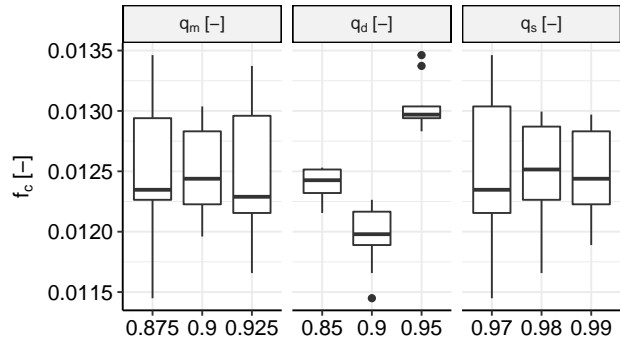

**Figure 11.** Sensitivity of $f_c$, which is the mean of the absolute differences in correlation between the synthetic and the observed descriptor sets for all locations, to the three main settings used for the statistical analysis in Sect.[4]. $q_m$ is the quantile threshold for the GPDs, $q_d$ is the quantile threshold for HT04, $q_s$ is the quantile threshold for the simulation from HT04.

### 5.2.2 Dependence structure

In Sect.[2.2], we stated the objective to generate a large catalogue containing synthetic discharge peaks, filling up the observed distributions while retaining the observed dependence structure (OBJ2). We defined what we considered two key features for the quality of the generated synthetic catalogue. First, it had to contain descriptions of a much larger variety of hypothetical

(synthetic) events than those identified in the observed data (KF1), which we achieved by sampling a large number of synthetic descriptors and transforming them to follow the same distributions as fitted to the observed. Second, the dependence structure of the synthetic catalogue needed to agree with that of the observed, since the observed descriptor catalogue had to be a likely subset of the synthetic (KF2), which we achieved with the dependence model, of which the results were demonstrated in Fig.(8)-Fig.(9). We further investigated the sensitivity of the results for KF2, using a summary descriptor of Fig.(8).

Fig.(11) shows the sensitivity of $f_c$, which is the mean absolute difference in Spearman correlation between the synthetic and the observed descriptor sets for all locations. No clear trend was found for both $q_m$, which is the quantile threshold for the GPDs, and $q_s$, which is the quantile threshold for the simulation from HT04. A local minimum was found for $q_d$, which is the quantile threshold to select the observed descriptors to which the HT04 model was fitted.

A recent, more comprehensive study of the sources of uncertainty in a probabilistic flood risk model was provided by Winter

et al. (2018), who used the framework provided by Hall and Solomatine (2008).

### 5.3 Applicability to pan-European FRA

The generated synthetic descriptor catalogue can be used to drive an event-based chain of models, which may cascade from a hydraulic model of the river network coupled with an inundation model to damage and/or life safety models. To drive an inundation model, synthetic discharge events have to be reconstructed from the synthetic descriptor sets in the catalogue,

which corresponds to what would be step 4 in Fig.(1), moving down from synthetic descriptors towards synthetic events. This

step comprises fitting discharge hydrographs to the synthetic descriptors and assigning time lags. Difficulty can be expected in that the synthetic descriptor sets partially consist of synthetic discharge peaks and partially consist of synthetic auxiliary values. For the synthetic peaks, hydrographs can be reconstructed by fitting a typical (triangular) hydrograph shape to the synthetic peaks, whereas for the synthetic auxiliary values, it will not be entirely clear how to fit a hydrograph. Time lags for
the synthetic descriptors sets can be borrowed from the corresponding observed descriptors sets, which would implicitly use the assumption that travel times, i.e. wave celerities, are independent of magnitude.

However, these difficulties are not specific to this analysis, but apply for all analyses in which an event-based approach is combined with descriptors per location. The catalogue of synthetic discharge event descriptors was provided at 298 locations for a synthetic period of 10.000 years (with stationary climate conditions). Both the number of locations and the number
of synthetic years can be expanded to provide a more detailed coverage. This catalogue will be used to generate discharge hydrographs to drive a pan-European inundation model for continental, event-based flood risk assessment.

## 6   Conclusions

We used a new 'noise removal' and 'wave tracking' method with which we successfully identified discharge waves in all major European river basins. Using global time windows, we clustered these river basin events to pan-European events. With
a mixture multivariate dependence model, we managed to capture the dependence structure between discharge peaks of daily discharge at 298 different locations on the river network of major European rivers. We created a catalogue of spatially coherent synthetic event descriptors, containing 10.000 years of synthetic discharge peaks with a dependence structure that is similar as in the observed data, thereby showing spatially coherence. This catalogue is a starting point for the exploration of the range of possible scenarios of pan-European flooding and associated probabilities, which is the foundation of flood risk assessment.

*Author contributions.*  DD did the research. He developed the main ideas brought forward in this paper and wrote the R-code to obtain results and figures. YL was involved in all aspects of the research, with a specific focus on the statistical section. He contributed significantly to the writing of this paper. BG devised the idea to apply the current methodology for joint discharge peaks to a continental domain. FD provided general feedback and in particular provided the method used to check the extremal dependence. SV provided general feedback, helped improve the manuscript's positioning and significantly helped clarify the applied methodology.

*Competing interests.*  The authors declare that they have no conflict of interest.

*Acknowledgements.*  This research is part of the 'System-Risk' project and has received funding from the European Union's Horizon 2020 research and innovation programme under the Marie Skłodowska-Curie grant agreement No 676027.

We thank the European Joint Research Centre for providing the European discharge data set.

We thank D. Paprotny, B. Jongman and H. Winsemius for their critical remarks, which increased the quality of the manuscript.

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
