# Peer review of "Stochastic generation of spatially coherent river discharge peaks for continental event-based flood risk assessment."

_Natural Hazards and Earth System Sciences, 2018_

## Short Comment (SC1) · 20 Aug 2018

At the request of the author I have copied a series of questions and answers from a private correspondence relating to the manuscript: Stochastic generation of spatially coherent river discharge peaks for large-scale, event-based flood risk assessment

Q1. You used 25 years of data to form the dependence. Do you think there are limitations from this? I.e. should we expect that there are many more types of events (footprints for instance) that will be missed and therefore, incapable of being represented in the model outputs? I often find people ask about this due to the interest in 'black swan' events.
A1. We go beyond observations by extrapolation of values only. We do not go beyond in terms of footprint, since the statistics are applied to particular locations. We are planning to vary (and extrapolate) spatio-temporal footprints. We may address this in my next paper (precipitation) or the one after (compound).

Q2. As I understand it, you use the NR method to find peaks at individual sites. You then split the 25-year timeseries into 21-day bins, essentially taking the largest peak (or max non-peak if no peak is present) from all sites during that bin to represent that 'event' peak. This provides you with a consistent number of events per year that is defined by your choice in bin size (21 days in this instance). Does this mean that there is no minimum magnitude used when defining an 'event' at a given site in the observed data?

A2. The NR method is based on fluctuations, so does not yield events with a minimum magnitude (as POT does). - See the next question (or the paper again) for clarification on the procedure.

Q3. Linked to the above – is it likely that within an event bin there might be 2 independent storm systems leading to flooding, and given the method, result in i. spurious correlations, and ii. an underestimate of the number of independent events taking place?

A3. Per location: - When we apply NR to the 298 locations, we get a different number of events at each location. So we cannot cast all peaks into a matrix. To match the peaks: - We apply NR to each pixel in the network. - We track discharge waves (river basin events). - To the EU events (time windows) we assign entire tracked discharge waves, based on the first time entry of the event (in which time window it falls). - Per EU event we only keep the entire tracked discharge wave with the largest discharge value (somewhere). Cheating: - From EU events, we extract the peaks at the 298 locations. This yielded an incomplete matrix (see Table 1). - This means that per column (i.e. per location), we have to fill gaps with 'auxiliary' (dummy) values and drop peaks when

there are multiple candidate peaks for an entry. - In my personal opinion (Dirk), unless statisticians come up with a way to handle incomplete matrices, this is a dead end for the combination of an event-based approach and descriptors (e.g. peaks) per location. Therefore, in my next analysis, I will not be working with descriptors per location.

Q4. You open with a discussion about the importance of antecedent conditions in flood event generation, but I am not entirely clear how that is represented in your model? If you do address this then you could really hammer this home as that seems to me to be something that these sorts of approaches often struggle with.

A4. We do not address antecedent conditions. Maybe we should drop the mention of it in the introduction, since we do not address it and we might be raising false hopes. Or maybe we shouldn't drop it, since I guess it is important enough to be mentioned. Let's see what the reviewers think.

Q5. Did you look at the role of seasonality in the dependence? Do you think it would make much difference to the results?

A5. - We do not address seasonality. - More generally, there is a full stationarity assumption, which can be challenged in many ways (seasonality, trends, human inter- ference in the system, etc.). - We only focussed on large-scale, spatial dependence.

Q6. RE: defining marginal distributions - Does infilling the 'missing peaks' have an impact on the return period estimation of gauge timeseries? If so do you have any idea of how significant it might be? Also, was there much variation in the optimal marginal threshold across sites, and if so, do you think applying an individual marginal threshold to each site would make much difference to your results?

A6. - It does indeed affect the marginal distributions, and thus the return levels. We could analyse this effect, but the marginal distributions are not the main focus of this work. - Since we did not focus on the marginal distributions we just applied the marginal threshold by quantile. It could definitely be improved by finding optimal thresholds per

location. - Just to mention, we filled the gaps to be able to capture the dependence structure. I think we would not be able to capture the (entire) dependence structure without filling the gaps. Therefore, the effect on the dependence structure would be very hard to analyse.

———————————————————

---

## Referee Comment (RC1) · D. Paprotny (Referee) · 28 Aug 2018

The study which I had the opportunity to review aims at producing a synthetic series of discharge peaks that preserves spatial dependence across various locations in Europe. The manuscript is an important contribution to solving the problem of generating pan-European events using statistical methods. The methodology is explained well and the paper is pleasant to read. However, the paper has one major drawback and a number of smaller points related to how the research is presented.

The main issue is that the study abruptly ends just when it gets most interesting. Most of the paper is covered by methodology (which is understandable given the scope of

the paper), then followed by a section on validation and suddenly ends with a very short conclusion. Therefore, the reader is left with a number of questions, that would be normally addressed in a proper results and discussion sections:

1. How do the results look like?

2. What is the uncertainty in the results?

3. What is the sensitivity to the choice of thresholds for the analysis?

4. How the resulting dataset can be applied to a pan-European flood risk assessment?

5. What are the next research steps?

Ad. 1: it would very beneficial to presents and analyse the resulting dataset, as it is interesting to know how large are the pan-European events? What is the total peak discharge of an average event, annual maxima, or a 1 in 100 years event? How different is the discharge computed with the authors' methodology from discharge computed independently for each station? Is there spatial variation within Europe how much the discharge differs between the assumptions of dependency between locations and of complete independence?

Ad. 2: using copulas has the benefit that the uncertainty can be easily obtained. Can this information be used to show what is the uncertainty in the discharge during a synthetic pan-European flood with a large return period, e.g. 1 in 100 years? Or at least the uncertainty in discharge for individual locations?

Ad. 3: The paper mentions selecting parameters through "trial and error" no less than four times. That includes parameters of the noise reduction, time window of the pan-European events, threshold parameters of GPD fitting and multivariate analysis. How sensitive are the results to the choice of parameters? How much would the discharge at a given location change if one of the parameters is modified, or the total discharge of a pan-European event with a given return period?

Ad. 4: The paper states in the title that the resulting dataset could be used for "large-scale, event-based flood risk assessment". However, it is rather unclear how would it be applicable to such studies. The authors only calculate the discharge for 298 locations, which is around 0.1% of all EFAS grid points, or less than 1% of flood zone calculation subdomains used by Alfieri et al. for a pan-European flood hazard assessment. Therefore, what is the utility of such a small number of locations of Europe-wide assessments? Is it possible to scale it up? I guess that might be a problem, as more locations means much more computational burden for the multivariate analysis. Is the 298 locations the maximum feasible number, or was chosen for quick testing of the method and can be easily increased?

Ad. 5: related to previous points, it is unclear from the papers what are the limitations of the method (if any) and what should be done to improve it. Also, how can future climate change be added to analysis? How can the result be applied to e.g. computation of flood hazard zones, annual expected losses in Europe or pan-European losses with a given return period?

I really don't expect the authors to tackle fully all my questions, but a proper discussion section in the paper is needed to make the reader familiar with the method's limitations, uncertainty, sensitivity, applicability to European-scale FRA and future research outlook. The aspect of application to FRA and overall the paper's contribution should also be an important point in the conclusions, which at present read more like a part of an abstract. Also, the questions posed by Niall Quinn in his short comment are good points that should be taken into consideration in a discussion section. Where necessary, the authors also might want to move some paragraphs from the methodology to the discussion (e.g. P8L8-15) to avoid overlaps and make the methodological sections more on point.

Minor points:

1. The abstract needs to be rewritten, as presently it consists almost entirely of background and seems to serve as a introduction to the whole paper. It should be a balanced representation of the rationale, methods, results and conclusions. It also shouldn't contain any references. I would suggest then to move the two first paragraphs of the abstract into the paper's introduction (as the literature review in the abstract is relevant but not repeated in the main text), and use the text from the conclusions to make a new abstract.

2. In section 2, the data selection is not explained. It is written that "the network was reduced to the major streams and tributaries", but no information what threshold was applied here. Next, it is written that "we selected 298 representative locations within the network of major European rivers", but no selection criteria is provided, making it particularly unclear why those locations are "representative" and do not include some regions of Europe (as per Fig. 1). The extra information might be included as a supplement so that it won't slow down the pace of the paper.

3. Figures in the manuscript need various improvements. Fig. 1 misses both a legend and a scale, including explanation what variable is shown in the background and what is the data source. Fig. 2 has a rather uninformative description "Framework". Fig. 3 need indication of panels (a,b,c), as they are mentioned in the text, and a legend. The Y axis label should be "discharge" rather than "flow" and the units should conform to the journal's guidelines. In Fig. 4 and 5 full axis labels should be used. In Fig. 6, the data source is not stated (which station is it?), and I have doubts whether the axis labels are correct – please check. Also, it might make a better visual impression if the font sizes were uniform.

4. The authors use "large-scale" many times, including the title, in the sense of analysis covering a large area. However, in cartography "large-scale" has opposite meaning to the common one, and refers to maps covering a small area. Therefore, I would suggest to change "large-scale" to "continental-scale" to avoid the confusion.

5. P2L7: the authors mention a rather loosely-worded definition of a flood, which is

neither strict nor of much relevance as the paper deals with river discharge peaks regardless whether they cause a flood or not. I think it's best to omit the first sentence of the paragraph.

6. P3L31: citation missing.

7. P6L9: "River discharge waves generally propagate through the network in downstream direction". I guess it should be "always" instead of "generally".

8. In the description of HT04, it is unclear which cited paper gives detail of the fitting. The authors also might consider providing a supplement with details of the technical aspects of the multivariate analysis.

I am looking forward to the authors' revision of this important paper.

---

## Referee Comment (RC2) · B. Jongman (Referee) · 25 Sep 2018

Thanks for the opportunity to review this interesting and well-written paper. The modelling of event dependencies on a continental level is indeed highly relevant both scientifically and for the purpose of insurance and flood management.

The authors present a sophisticated approach to address this problem. However, the paper has several major issues that should be addressed before it can be published. The main issues are (1) the definition and validation of events; (2) the limited and highly technical description of data, modelling steps and results, which seem at times a high abstraction from the stated purpose of the paper (i.e. flood event analysis for

risk assessment); (3) the description of the implications, uncertainty and potential use of the results; and (3) the limited reference and comparison to previous studies in this field. Please see more detailed comments below.

Abstract: The abstract is unconventional with many references. It also does not clearly state the problem (why is the event-scale analysis needed?) and the implication/use of the results (it stops at outputs). I suggest to review and revise accordingly. The references should be better incorporated in the main body of the paper to address the lack of integration in literature (see other comments).

Context: the Abstract clearly puts this paper in line with current continental-scale flood modelling work that is based on statistical return period analysis rather than event identification (Ward et al., 2013; Alfieri et al., 2014; Dottori et al., 2016; Vousdoukas, 2016;Winsemius et al., 2016; Paprotny et al., 2017). However, the Introduction does not mention these previous modelling exercises and how this paper will differ/improve on those. I suggest to revise the introduction to make this context in existing literature stronger. Similarly, in the main text of the paper, the authors fail to compare their methods and results to other scientific work. For example, how do the correlations compare to those showed in previous studies (e.g. Jongman et al (2014); Timonina et al 2015, Risk Analysis), some of which are mentioned in the abstract but at no other stage in the paper?

Events: The definition of an event is indeed challenging, as explained in section 1.2.1 and 1.2.2. However, I feel the authors should use a consistent definition and use of the term 'event' (or various definitions thereof). In section 1.2.2 the authors introduce the use of 'block' and 'dynamic' events, but the later chapters mainly use 'local', 'pan-European', 'atmospheric' events. Indeed, the word 'block' and 'dynamic' are not featured beyond section 1! See also the next two comments.

River basin events: In the description of the computation of river basin events I miss a critical discussion of the hydrological/hydraulic model. Since the analysis is conducted

using a model instead of observation data, the upstream-downstream correlations (i.e. river basin events) are obviously more strongly dependent on the model than on reality. How are the correlations affected by this model and what are the implications? Is it even realistic to do this, or does such upstream-downstream analysis simply result in backward-engineering of the discharge model? This should be incorporated in section 3.2 and result discussion.

Pan-European events (section 3.3, Figure 5): I have an issue with the interpretation of the concept of pan-European events by the authors. The way I understand it from the methodology, Pan-European events are simply identified by finding one local peak and opening a 21-time window, during which all high discharge values across Europe are incorporated. However, simultaneous high discharge does not make an 'event'. Where does the geographical boundary of an event stop? Would simultaneous high values in Germany and China be considered an event? Or two high rainfall events in one month caused by two separate low pressure systems in different parts of Europe? In my opinion it would be important to incorporate a concept of correlation or at least the identification of a single weather/atmospheric pattern that links the discharge values, to eliminate individual coincidental simultaneous occurrences (i.e. after Section 4).

Multivariate model: This section is extremely cryptic and very hard to read or relate to the rest of the paper. While trying to relate the model to the identification of events based on discharge (which I presume is the aim), one finds that the words 'event' and 'discharge' are not mentioned a single time in the entire section 4 (apart from once in the figure caption)! It is currently very hard to understand what the inputs, processing and outputs are. I suggest to revise and align it strongly with Section 3.

Validation: it is not clear to me what 'synthetic' and 'observed' data is used for validation. It would be important to validate the outcomes of the multivariate model with fully independent datasets, i.e. actual event data (preferably); recorded atmospheric patterns; observed river discharge data; etc. Currently it seems much of the validation is done within the model, which would make it very hard to de-couple the internal workings of the hydrological model from the results (e.g. the strength of intra-basin correlations are a given, since the hydrological model is the basis of the analysis). Similarly, there is no discussion on potential regional and temporal differences in model precision (for example due to weather patterns, snow melt, data availability, river size, and other real-world aspects). As such, the validation, just as the model description, remains highly abstract. This section needs substantial strengthening and, once again, needs to actually reflect on 'events' to prove that it does what the paper aims to do.

Results and use: The paper abruptly ends after the technical multivariate model section, and the conclusion section is only a few lines. It remains unclear to the reader what the exact result is of this study and how it can be used. In the abstract, the authors claim that their study is needed to analyze insurance portfolios. Can it? How do we use the model for probabilistic risk assessment? How would that be better than currently available models, and what are the uncertainties? Which regions does it work and which does it not? It would strongly benefit the paper to add a section on the resulting data, its applications and limitations.

---

## Author Comment (AC1) · 16 Nov 2018

Summary We would like to start with thanking mr Paprotny for his thorough and constructive review and apologise for the slight delay in reply.

First, we will address his points one by one. Second, we will provide a list with proposed improvements. We would be happy to hear from the reviewer, after reading the comments to his points, if we have missed/neglected/misinterpreted any major points.

Referee (D. Paprotny), main 1. How do the results look like? Ad. 1: it would very beneficial to presents and analyse the resulting dataset, as it is interesting to know how large

are the pan-European events? What is the total peak discharge of an average event, annual maxima, or a 1 in 100 years event? How different is the discharge computed with the authors' methodology from discharge computed independently for each station? Is there spatial variation within Europe how much the discharge differs between the assumptions of dependency between locations and of complete independence?

Response (Diederen et al.): This paper expands the methodology of multi-site analyses to large-scale. As usual in multi-site analyses, the (tail-end) General Pareto Distributions (GPDs) connect magnitudes to probabilities (and therefore to return periods), which are captured locally (per location). These distributions are not the main focus of this study. We would like to point to p12,l9: "After the simulation, we transformed the margins of the synthetic 10 descriptor matrix to respect the fitted GPDs, thereby slightly distorting the dependence structure." This implies that, with this methodology, the discharge peaks computed with the dependence model are exactly the same as when the discharge peaks would be calculated independently (directly drawn from each individual distribution). In short, magnitudes and probabilities depend completely on the fits of the local GPDs, for which we applied a very simple methodology (Maximum likelihood, fitted to data above a fixed quantile), which we think is not too interesting to present. The main result in this paper is the capturing (Sect.3) and modelling/reproducing (Sect.4) of the spatial dependence structure. The captured and reproduced spatial dependence structure is displayed in Fig.8. (for the entire joint distributions) and figure 9 (specifically for joint tail-ends of the distributions).

Referee (D. Paprotny), main 2. What is the uncertainty in the results? Ad. 2: using copulas has the benefit that the uncertainty can be easily obtained. Can this information be used to show what is the uncertainty in the discharge during a synthetic pan-European flood with a large return period, e.g. 1 in 100 years? Or at least the uncertainty in discharge for individual locations?

Response (Diederen et al.): The notion of return period has no specific meaning within a multi-variate framework. In each individual set of peaks (each row that describes an

event), each peak (at a different location) has a different exceedance probability (and therefore a different return period). Therefore, no return period can be assigned to a set of peaks. The uncertainty at individual locations could be investigated, which would imply investigating the GPD fits, which, as previously mentioned, is not the focus of this study.

However, we agree that uncertainty should be discussed and brought forward. We will provide better referencing to relevant recent studies that specifically focussed on uncertainty:

Hall, J. and Solomatine, D.: A framework for uncertainty analysis in flood risk management decisions Winter, B., Schneeberger, K., Huttenlau, M., and Stötter, J.: Sources of uncertainty in a probabilistic flood risk model)

Referee (D. Paprotny), main 3. What is the sensitivity to the choice of thresholds for the analysis? Ad. 3: The paper mentions selecting parameters through "trial and error" no less than four times. That includes parameters of the noise reduction, time window of the panEuropean events, threshold parameters of GPD fitting and multivariate analysis. How sensitive are the results to the choice of parameters? How much would the discharge at a given location change if one of the parameters is modified, or the total discharge of a pan-European event with a given return period?

Response (Diederen et al.):

This is a very fair and important comment. Unfortunately, We have explored a couple of settings for each of the parameters mentioned, which classifies as trial and error. If necessary, we could explore a couple more settings and report on the difference between outcomes using different settings. However, because of the complex methodology applied to a large data set (long computational times), a full sensitivity analysis would not be feasible.

With regards to the settings of the statistical model, a recent study specifically addressed the settings using in the HT04 model: Winter, B., Schneeberger, K., Huttenlau, M., and Stötter, J.: Sources of uncertainty in a probabilistic flood risk model)

We suggest that the sensitivity of the methodology should be further tested in future research.

Referee (D. Paprotny), main 4. How the resulting dataset can be applied to a pan-European flood risk assessment? Ad. 4: The paper states in the title that the resulting dataset could be used for "largescale, event-based flood risk assessment". However, it is rather unclear how would it be applicable to such studies. The authors only calculate the discharge for 298 locations, which is around 0.1% of all EFAS grid points, or less than 1% of flood zone calculation subdomains used by Alfieri et al. for a pan-European flood hazard assessment. Therefore, what is the utility of such a small number of locations of Europe-wide assessments? Is it possible to scale it up? I guess that might be a problem, as more locations means much more computational burden for the multivariate analysis. Is the 298 locations the maximum feasible number, or was chosen for quick testing of the method and can be easily increased?

Response (Diederen et al.): As part of the system-risk project (https://system-risk.eu), the task of the reconstruction of (spatially-coherent) hydrographs lies with the partner in Bristol, who do large-scale inundation simulations (Lisflood) and requested this set of synthetic discharge peaks (at these specific locations). To drive their inundation models, they will use the peaks to set up discharge boundary conditions (hydrographs), i.e. they do not require discharge peaks at each grid point. Alfieri et al. only address distributions per grid cell, which is something that can be done more easily on a high resolution. Addressing the dependence structure means that including each location gives an additional dimension (column) to the multivariate matrix. The number of locations was indeed for testing and could be easily expanded, with mainly the drawback of (again) longer computational times. However, it cannot be expanded to extremely large numbers of locations (high resolution grids) because of the dimensionality of the multivariate analysis.

Referee (D. Paprotny) What are the next research steps? Ad. 5: related to previous points, it is unclear from the papers what are the limitations of the method (if any) and what should be done to improve it. Also, how can future climate change be added to analysis? How can the result be applied to e.g. computation of flood hazard zones, annual expected losses in Europe or pan-European losses with a given return period?

Response (Diederen et al.): The main limitation of this method is discussed in 1.2.3 Handling dynamic events in a statistical event generator (p3, l1). This limitation is brought forward and discussed in the manuscript, because it is a limitation that does not specifically apply to this analysis only, but it applies to all large-scale, multi-site, event-based analyses. This type of analysis requires peaks (or other descriptors) at specific locations, whereas, within a large-scale framework, events do not produce peaks at all locations. This limitation implies that multi-site analyses will always require trade-offs (for the main trade-off of this study, see p8,l9). Climate change is not addressed in this study. It would require an expanded methodology (somehow incorporating trend analysis and dependent sampling within a multi-site framework) and would require (again) a lot of additional computational power and effort. Including trend patterns would reduce the quality of the spatial dependence structure, since such a methodology would lead to (statistical) trade-offs. Flood hazard zones and annual expected losses could be calculated after using these boundary conditions to drive a European-wide inundation model which then, subsequently, would be used to drive damage models. This study focusses on the generation of the boundary conditions, not on the input-output models belonging to the flood risk modelling chain.

Referee (D. Paprotny), minor 2. In section 2, the data selection is not explained. It is written that "the network was reduced to the major streams and tributaries", but no information what threshold was applied here. Next, it is written that "we selected 298 representative locations within the network of major European rivers", but no selection criteria is provided, making it particularly unclear why those locations are "representative" and do not include some regions of Europe (as per Fig. 1). The extra information

might be included as a supplement so that it won't slow down the pace of the paper.

Response (Diederen et al.): Although significant effort was put into finding the river network in the data set (using the mean flow per pixel and a river finding algorithm), we do not consider this part of the analysis important for publication, since the location of European rivers is well known, especially of the major ones. All that would theoretically be required is access to a database containing this information. The 298 locations were selected with the criterion of decent coverage of the river network. Other locations could be used, but this would not change the methodology, which we focus on in this paper.

Referee (D. Paprotny), minor 5. P2L7: the authors mention a rather loosely-worded definition of a flood, which is neither strict nor of much relevance as the paper deals with river discharge peaks regardless whether they cause a flood or not. I think it's best to omit the first sentence of the paragraph.

Response (Diederen et al.): We agree and will omit the sentence.

Proposed improvements 1. We will improve the legends and scales in the figures (where required). 2. We will better reference to studies that consider uncertainty and sensitivity. We will add a discussion section as suggested, in which we will discuss uncertainty, sensitivity, applicability to European-scale FRA and future research outlook and we will see if we can move some discussion from the methodology section. In this section, we hope to address the reviewer's main points/objections [point 2 and 3] 3. We will see if we can provide better reference to the general HT04 fitting procedure. 4. We will completely revise the abstract as suggested [minor 1].

---

## Author Comment (AC2) · 16 Nov 2018

Summary We would like to start with thanking mr Jongman for his thorough and constructive review and apologise for the slight delay in reply.

A number of constructive comments was provided related to clarification and discussion of work presented in this manuscript. We are very happy to incorporate these and think it will improve the quality of the manuscript. A request was made to provide a quality check of the used (modelled) discharge data set. Unfortunately, this would be very hard for us to do (since we are not in possession of the model), so we feel this is out of scope and refer to the authors who generated the modelled discharge data.

[Figure]

First, we will address his points one by one. Second, we will provide a list with proposed improvements. We would be happy to hear from the reviewer, after reading the comments to his points, if we have missed/neglected/misinterpreted any major points.

Referee (B. Jongman), 1. Abstract: The abstract is unconventional with many references. It also does not clearly state the problem (why is the event-scale analysis needed?) and the implication/use of the results (it stops at outputs). I suggest to review and revise accordingly. The references should be better incorporated in the main body of the paper to address the lack of integration in literature (see other comments).

Response (Diederen et al.): This point was also made by reviewer one and we fully agree. We concluded that the abstract has not pointed readers in the right direction. We will completely revise the abstract to make it clear what the focus of this manuscript is.

Referee (B. Jongman), 2. Context: the Abstract clearly puts this paper in line with current continental-scale flood modelling work that is based on statistical return period analysis rather than event identification (Ward et al., 2013; Alfieri et al., 2014; Dottori et al., 2016; Vousdoukas, 2016;Winsemius et al., 2016; Paprotny et al., 2017). However, the Introduction does not mention these previous modelling exercises and how this paper will differ/improve on those. I suggest to revise the introduction to make this context in existing literature stronger. Similarly, in the main text of the paper, the authors fail to compare their methods and results to other scientific work. For example, how do the correlations compare to those showed in previous studies (e.g. Jongman et al (2014); Timonina et al 2015, Risk Analysis), some of which are mentioned in the abstract but at no other stage in the paper?

Response (Diederen et al.): In this manuscript we focus on the dependence structure.

Ward et al., 2013; Alfieri et al., 2014; Dottori et al., 2016; Paprotny et al., 2017; focus on local distributions (of modelled data, per grid-cell). Vousdoukas, 2016 focus on coastal variables. Paprotny et al., 2017 focus on a different type of dependence, which

is related to flood inducing factors (causal, instead of a hydrological model). Jongman et al (2014) do capture correlations of monthly discharge maxima in Europe. They use them to split Europe into different zones in which the correlation is relatively high. However, we look at events in more detail (building them up from tracked discharge waves in different river basins) and are more narrow in our framework (we do not address the entire flood risk modelling chain nor touch upon climate change). We captured the spatial dependence of discharge peaks and do not so much aim to analyse the obtained dependence (e.g. using spearman correlations), but aim to capture the dependence in the best we way can (wave tracking) and reproduce the dependence while generating a large synthetic data set (for which we use the spearman correlation). Timonina et al (2015) seem to focus on the dependence of flood losses, rather than on the dependence of discharge.

We conclude that the aims of these studies are quite different from ours and do not overlap in terms of methodology. The binding factor is the large spatial scale. We appreciate the referee's suggestions for additional references and we will include a discussion on the differences in methodology.

Referee (B. Jongman), 3. Events: The definition of an event is indeed challenging, as explained in section 1.2.1 and 1.2.2. However, I feel the authors should use a consistent definition and use of the term 'event' (or various definitions thereof). In section 1.2.2 the authors introduce the use of 'block' and 'dynamic' events, but the later chapters mainly use 'local', 'pan-European', 'atmospheric' events. Indeed, the word 'block' and 'dynamic' are not featured beyond section 1! See also the next two comments.

Response (Diederen et al.): The distinction between block events and dynamic events are used to place the studies into context. From the introduction onwards, all events referred to are dynamic events. We will clarify this in our revised manuscript.

Referee (B. Jongman), 4. River basin events: In the description of the computation of

river basin events I miss a critical discussion of the hydrological/hydraulic model. Since the analysis is conducted using a model instead of observation data, the upstream-downstream correlations (i.e. river basin events) are obviously more strongly dependent on the model than on reality. How are the correlations affected by this model and what are the implications? Is it even realistic to do this, or does such upstream-downstream analysis simply result in backward-engineering of the discharge model? This should be incorporated in section 3.2 and result discussion.

Response (Diederen et al.): No hydrological nor hydraulic model is used in this study. For a discussion of the quality of the used data set we would refer to the JRC, who produced the data set (Alfieri et al 2014). We agree that the dependence in the modelled data may/will differ from the dependence in reality. However, since we do not have the data of reality, it is impossible to compare, such that we would only be able to speculate. Moreover, we focus on a methodology to capture the dependence structure and retain it while generating a large synthetic data set. In the future, it can be readily applied to data sets that capture reality more closely. If we were to speculate, we would be fairly confident that waves can be tracked in reality and are looking forward to studies that apply wave tracking to data of the (relatively dense) European network of discharge gauges.

We will add in section 2 that the data is model output and, hence, not perfect, but that this is no problem as the focus is on the methodology. 

Referee (B. Jongman), 5. Pan-European events (section 3.3, Figure 5): I have an issue with the interpretation of the concept of pan-European events by the authors. The way I understand it from the methodology, Pan-European events are simply identified by finding one local peak and opening a 21-time window, during which all high discharge values across Europe are incorporated. However, simultaneous high discharge does not make an 'event'. Where does the geographical boundary of an event stop? Would simultaneous high values in Germany and China be considered an event? Or two high rainfall events in one month caused by two separate low pressure systems in different

parts of Europe? In my opinion it would be important to incorporate a concept of correlation or at least the identification of a single weather/atmospheric pattern that links the discharge values, to eliminate individual coincidental simultaneous occurrences (i.e. after Section 4).

Response (Diederen et al.): a) Correlation of discharge across different river basins can be expected, as a result of atmospheric systems spanning multiple river basins. To use atmospheric systems for event identification would require the usage of an atmospheric data set (e.g. precipitation instead of discharge for event identification). This will introduce a new set of problems to deal with (e.g. where a single 'atmospheric event' triggers multiple 'discharge events' or the other way around). To consider atmospheric data sets is out of scope for this study.

b) We would argue that there is no strictly correct way of defining an event, but that it is subjective what an event is. A large number of previous studies exists that use an event identification without spatial delineation (time window methods/'concurrent' or 'simultaneous' methods). For multi-site studies that use event identification methods without spatial delineation, the spatial domain is determined by the sites considered in the study.

c) We agree with the referee that, in the context of the large-scale studies, it becomes evident that event definition without geographical boundaries is not necessarily meaningful in the physical sense.

d) Discharge waves do not span across multiple river basins (by definition). However, atmospheric systems do (point a.). Large systems may be of the order of magnitude of the European continent. This study aims to capture the expected spatial correlation in river discharge as a result of atmospheric systems, even though the events may not physically make sense regarding the discharge.

e.) In summary, to capture spatial correlations across multiple sites may limit the physical meaningfulness of events, which becomes more evident with a larger scale. We

discuss this problem in Sect. 1.2.3 "Handling dynamic events in a statistical event generator", since the choice of a multi-site approach is related to the statistical method available. We will expand the discussion of this issue and provide recommendations for follow up studies of how to deal it.

Referee (B. Jongman), 6. Multivariate model: This section is extremely cryptic and very hard to read or relate to the rest of the paper. While trying to relate the model to the identification of events based on discharge (which I presume is the aim), one finds that the words 'event' and 'discharge' are not mentioned a single time in the entire section 4 (apart from once in the figure caption)! It is currently very hard to understand what the inputs, processing and outputs are. I suggest to revise and align it strongly with Section 3.

Response (Diederen et al.): In section 4 we are applying multivariate statistics to a matrix. A step is made from the physical world to the world of statistics, which has its own jargon. We will try to clarify the transition in Sect 3.4 in the revised manuscript.

Referee (B. Jongman), 7. Validation: it is not clear to me what 'synthetic' and 'observed' data is used for validation. It would be important to validate the outcomes of the multivariate model with fully independent datasets, i.e. actual event data (preferably); recorded atmospheric patterns; observed river discharge data; etc. Currently it seems much of the validation is done within the model, which would make it very hard to de-couple the internal workings of the hydrological model from the results (e.g. the strength of intra-basin correlations are a given, since the hydrological model is the basis of the analysis). Similarly, there is no discussion on potential regional and temporal differences in model precision (for example due to weather patterns, snow melt, data availability, river size, and other real-world aspects). As such, the validation, just as the model description, remains highly abstract. This section needs substantial strengthening and, once again, needs to actually reflect on 'events' to prove that it does what the paper aims to do.

Response (Diederen et al.): The paper is about a methodology to generate a large catalogue of synthetic events that captures the spatial variability found in a short historical time series. Since the focus is on the methodology, we find it acceptable that this is done with modelled data.

We will change the name of the 'validation' section to ' validation of statistical model', and we will emphasise in the text that this is not a validation of the resulting data set, but only shows that the observed is a likely subset of the synthetic with respect to the dependence structure.

Referee (B. Jongman), 8. Results and use: The paper abruptly ends after the technical multivariate model section, and the conclusion section is only a few lines. It remains unclear to the reader what the exact result is of this study and how it can be used. In the abstract, the authors claim that their study is needed to analyze insurance portfolios. Can it? How do we use the model for probabilistic risk assessment? How would that be better than currently available models, and what are the uncertainties? Which regions does it work and which does it not? It would strongly benefit the paper to add a section on the resulting data, its applications and limitations.

Response (Diederen et al.): In this manuscript we present a method to generate synthetic data, which is to be used to force the flood risk model chain. This step is decomposed into two main sub-steps (event identification and description, generation of synthetic peaks), both of which are challenging.

It is different from other studies in that the result is a catalogue of large-scale, synthetic events of discharge peaks which spans across Europe, which incorporates the spatial dependence structure, i.e. are spatially coherent. Such a data set did not exist previously (to the best knowledge of the authors).

To analyse insurance portfolios: 1) discharge hydrographs have to be reconstructed. 2) a European-wide inundation model has to be run (in event-based mode). 3) Inundation data has to be converted to damage/losses using a flood loss model. The result would
be a synthetic catalogue of European-wide events of flood damage (to be used by planners, e.g. (re)-insurance). Follow up work (at least steps 1 and 2) is expected from the Bristol group (Bates et al) within the context of the System-Risk project.

We will elaborate these points in the text and emphasise that it is only the first step in a chain of steps towards a synthetic catalogue of European-wide events of flood damage.

Proposed improvements

a) We will completely revise the abstract as suggested [point 1]. b) We will add references and improve the study's positioning in the introduction [point 2]. c) We will try to provide more clear definitions of the event-type used in this study [point 3]. d) We will aim to improve Sect.1.2.3 "Handling dynamic events in a statistical event generator", in which we discuss the main limitation of the methodology applied in this study (events without spatial delineation) raised by the referee [point 5]. e) We will try to clarify Sect.3.4, where the transition is made from physical jargon to statistical jargon [point 6]. f) We will change the name of the 'validation' section to 'model validation', and we will emphasise in the text that this is not a validation of the resulting data set, but only shows that the observed is a likely subset of the synthetic with respect to the dependence structure [point 7]. g) We will better describe how the resulting data set is to be used in practice [point 8].

---

## Author Response (AR1)

**Stochastic generation of spatially coherent river discharge peaks for  continental-scale event-based flood risk assessment.**

**Dirk Diederen[1], Ye Liu[1], Ben Gouldby[1], Ferdinand Diermanse[2], and Sergiy Vorogushyn[3]**

[1]HR Wallingford, Crowmarsh Gifford, United Kingdom
[2]Deltares, Delft, Netherlands
[3]GFZ German Research Centre for Geosciences, Potsdam, Germany

*Correspondence to:* Dirk Diederen (dirkdiederen@hotmail.com)

Compiled: February 22, 2019

**1 Point-by-point response**

**1.1 Referee 1**

For the point-by-point response to mr Paprotny, please see nhess-2018-231-RC1.

**1.2 Referee 2**

For the point-by-point response to mr Jongman, please see nhess-2018-231-RC2.

**1.3 Editor**

First of all we thank you for your comments that will help us improve our paper. In response to your final editor report, we would like to discuss our interpretation of the points you have raised and consult with you whether you think our intended additional analysis will be sufficient.

*1. In particular the notion of assessing uncertainty has not been addressed to my satisfaction. The authors say a number of times that it is difficult to assess uncertainty, or validate, and that using model output is justifiable because the focus of the paper is the method, rather than results. However, for a proper justification that the method is valid, an assessment of its uncertainty and/or validation against some real data (synthetic events against actual events from observations as suggested by Jongman) is required. The lowest hanging fruit to do this to my mind, is to benchmark the method against a set of observations, and perform assessment of the robustness of the method by testing a number of assumptions (for instance the GPD quantile threshold). This will require some additional work.*

We think this is a fair point and propose the following amendment to assess uncertainty. We will explore different settings for the three parameters used for event identification/description, to explore the parameter space around the settings we chose. Limited by long computational times, we think it is feasible to provide the results of 3*3*3=27 different parameter choices, thereby exploring the sensitivity of parameter choices. With regards to the uncertainty of parameter choice in the statistical model, we found a recent study entirely dedicated to this aspect https://rd.springer.com/article/10.1007/s11069-017-3135-5, and hope that providing reference to this article will be sufficient. We think that the synthetic data cannot directly be compared against observations at gauging stations, since it comprises hypothetical scenarios that have not occurred. The observed data has to be a likely subset of the synthetic, which is a point that we will better clarify in the revision by more accurately stating the objectives of this study.

*2. Compare against work by previous authors as suggested by Jongman. Even if this is qualitatively done, for instance by comparing a spatial plot of outcoming correlations of events against correlations found by Jongman et al., 2015 over a certain area would give a reasonable idea if the methods show the same spatial correlation patterns. I leave it to the authors to find a method to compare.*

We think this is a good idea and are prepared to compare our results to those of Jongman (2014). We won't be able to produce the exact same figures (not for all pixels), but will be able to produce spatial dependence figures entailing the 298 chosen representative locations, which we think should be sufficient for comparison.

*3. The comment on the used hydrologic/hydraulic model by Jongman needs to be better elaborated upon. They claim that "no hydrological/hydraulic model has been used for this study", but they do depend strongly on results by Alfieri et al. 2014, that are generated using a hydrology/hydraulics model cascade. The authors should explain what this entails, and discuss if and how the schematization can result in correlations that have no relation with reality but are a result of the schematization. Further "we do not have data of reality" may also not be true. There will certainly be basins in the study region where daily discharge data can easily be obtained by the authors.*

We will better clarify on which aspect we focus on specifically, and will better reference to the source of the modelled data used as input in this study. We do think that comparison with time series of local discharge measurements is outside the scope of this study (which takes the modelled data as input). We will, however, summarise the findings of the study in which the modelled discharge data was generated, in particular their conclusions on the comparison between directly observed and modelled discharge data.

*4. the missing fields author contributions, competing interests and disclaimer have to be treated.* We will fill out these missing fields (we thought that they were provided as "optional" in the latex template).

**2   List of relevant changes**

We have:

1. completely rewritten the abstract,

2. tried to improve the positioning in the introduction,

3. more clearly stated the objectives, framework and quality checks,

4. tried to clarify the different types of events discussed,

5. expanded the comparison with other literature in section 5.4,

6. moved parts to the discussion section as suggested,

7. expanded the discussion with a sensitivity analysis, for which we had to do a lot extra computation and rewriting of scripts,

8. added a section on the applicability of the study,

9. improved the quality of the figures as suggested,

10. revised the conclusions.

**Abstract.** ~~Flood risk assessments are required for long-term planning, e.g. for investments in infrastructure and other urban capital. Vorogushyn et al. (2018) call for new methods in large-scale 'Flood Risk Assessment' (FRA) to enable the capturing of system interactions and feedbacks. With the increase of computational power, large-scale, continental FRAs have recently become feasible (Ward et al., 2013; Alfieri et al., 2014; Dottori et al., 2016; Vousdoukas, 2016, .~~ We present a new method to generate spatially coherent river discharge peaks over multiple river basins, which can be used for event-based probabilistic flood risk assessment on a continental-scale. We first extract extreme events from river discharge time series data over a large set of locations by applying new peak-identification and peak-matching methods. Then we describe these events using the discharge peak at each location, whilst accounting for the fact that the events do not affect all locations. Lastly we fit the state-of-the-art multivariate extreme value distribution to the discharge peaks, and generate from the fitted model a large set of spatially coherent synthetic events. We demonstrate the capability of this approach in capturing the statistical dependence over all considered locations. We also discuss the limitations of this approach and investigate the sensitivity of the outcome to various model parameters.

*Copyright statement.* The author's copyright for this publication is transferred to HR Wallingford, Deltares and GfZ.

[revised manuscript text omitted]

---

## Referee Report (RR1)

Dear authors, dear editor

Thanks for the opportunity to review this interesting paper in revised form.

The authors did a good job in revising the papers based on reviewers' comments. The new paper is more clear in its scope, aims, limitations and applications.

I found the answers to my questions somewhat easy in places. The authors often rejected the suggestions for comparisons with other methods or validation material. I am quite concerned by argumentation that "since we do not have the data of reality, it is impossible to compare" and "the data is model output and, hence, not perfect, but that this is no problem as the focus is on the methodology". I am glad that these points were raised again by the editor and were addressed in the revised manuscript.

I am happy for the revised manuscript to be published in current form.

Best
Brenden

---

## Author Response (AR2)

**Stochastic generation of spatially coherent river discharge peaks for  continental event-based flood risk assessment.**

Dirk Diederen[1], Ye Liu[1], Ben Gouldby[1], Ferdinand Diermanse[2], and Sergiy Vorogushyn[3]

[1]HR Wallingford, Crowmarsh Gifford, United Kingdom
[2]Deltares, Delft, Netherlands
[3]GFZ German Research Centre for Geosciences, Potsdam, Germany

*Correspondence to:* Dirk Diederen (dirkdiederen@hotmail.com)

**1 Author's response**

*1. The conclusions and the new section 5.3 in my feeling still do not make a strong case for the study. It should be better explained why a 10,000-year synthetic series of very limited coverage is still preferable to a 25-year discharge series for whole of Europe. Currently section 5.3 mostly lists issues with applying the model, rather than making a positive case.*

We have expanded this section with a more positive note and have added to the conclusion that the synthetic catalogue is the starting point for event-based risk assessment (whereas the 25 years is only that which is observed).

*2. Figure 2 still misses both a legend and a scale, including explanation what variable is shown in the background and what is the data source. Similarly other figures miss legends and/or scale. In Figure 7 the positioning of panels doesn't seem to conform with the figure caption – please check.*

Figure 2 was revised. Legends, captions and labels have been adjusted.

*3. The authors changed the confusing "large-scale" in the title to "continental-scale", but expressions like "large-scale", "The larger the spatial scale" or "large spatial scales" etc., was retained throughout the paper. Please make sure this is corrected in the text.*

Scale is no longer mentioned in the manuscript (except in the references).

*4. "River discharge waves generally propagate through the network in downstream direction". I guess it should be "always" instead of "generally". (P6L11)*

"generally" was removed.

**Abstract.** We present a new method to generate spatially coherent river discharge peaks over multiple river basins, which can be used for continental event-based probabilistic flood risk assessment . We first extract extreme events from river discharge time series data over a large set of locations by applying new peak-identification and peak-matching methods. Then we describe these events using the discharge peak at each location, whilst accounting for the fact that the events do not affect all locations. Lastly we fit the state-of-the-art multivariate extreme value distribution to the discharge peaks, and generate from the fitted model a large  catalogue of spatially coherent synthetic  event descriptors. We demonstrate the capability of this approach in capturing the statistical dependence over all considered locations. We also discuss the limitations of this approach and investigate the sensitivity of the outcome to various model parameters.

*Copyright statement.* The author's copyright for this publication is transferred to HR Wallingford, Deltares and GFZ.

[revised manuscript text omitted]